# Orientation distributions of vacuum-deposited organic emitters revealed by single-molecule microscopy

Francisco Tenopala-Carmona [1,2] ✉, Dirk Hertel [1], Sabina Hillebrandt [1], Andreas Mischok [1], Arko Graf[2], Philipp Weitkamp[1], Klaus Meerholz [1] & Malte C. Gather [1,2] ✉

The orientation of luminescent molecules in organic light-emitting diodes strongly influences device performance. However, our understanding of the factors controlling emitter orientation is limited as current measurements only provide ensemble-averaged orientation values. Here, we use single-molecule imaging to measure the transition dipole orientation of individual emitter molecules in a state-of-the-art thermally evaporated host and thereby obtain complete orientation distributions of the hyperfluorescence-terminal emitter C545T. We achieve this by realizing ultra-low doping concentrations ($10^{-6}$ wt%) of C545T and minimising background levels to reliably measure its photo-luminescence. This approach yields the orientation distributions of >1000 individual emitter molecules in a system relevant to vacuum-processed devices. Analysis of solution- and vacuum-processed systems reveals that the orientation distributions strongly depend on the nanoscale environment of the emitter. This work opens the door to attaining unprecedented information on the factors that determine emitter orientation in current and future material systems for organic light-emitting devices.

Organic light-emitting diodes (OLEDs) are an important class of organic semiconductor devices that provide versatile, efficient emission in a wide range of applications. Improving device efficiency and stability by maximising the extraction of light from OLEDs is one of the main research priorities in this field[1,2]. Horizontal alignment of the transition dipole moment (TDM) of emitter molecules is a popular and impactful strategy to achieve this without needing to add a dedicated outcoupling structure, which can increase processing costs[3–6]. Instead, this strategy relies on making use of the anisotropic distribution of the dipole radiation from molecules that align during their deposition.

The emissive layers of efficient OLEDs are commonly fabricated by either thermally evaporating the molecules under high vacuum or depositing them from solution via spin-coating or printing. Several important factors that appear to promote horizontal TDM alignment in vacuum-processed emitters have been identified[3–9], and the alignment of small-molecule emitters in solution-processed layers for OLEDs is also receiving increasing attention[10,11]. However, many factors relevant to both processing methods—such as dipolar interactions, hydrogen bonding, host alignment, and the influence of the underlying layers—have only recently started to be examined systematically or remain unexplored[3,4,12–14]. In most state-of-the-art devices, small emitter molecules that efficiently convert electronic excitation into light are doped into host molecules to avoid the detrimental effects that frequently arise when two excited molecules are in close proximity[15,16]. Therefore, host-emitter interactions during film formation are particularly relevant to the orientation of emitters in doped films[3–6,12,14].

Our understanding of TDM alignment in these systems is limited by the amount of information we can obtain using current

[1]Humboldt Centre for Nano- and Biophotonics and Institute of Physical Chemistry, Department of Chemistry, University of Cologne, Köln, Germany. [2]Organic Semiconductor Centre, SUPA School of Physics and Astronomy, University of St Andrews, St Andrews, UK. ✉e-mail: F.TenopalaCarmona@uni-koeln.de; Malte.Gather@uni-koeln.de

measurement techniques. Most molecular-orientation studies rely on angle-resolved photoluminescence spectroscopy (ARPL)−or related angle-resolved luminescence spectroscopy techniques−and variable-angle spectroscopic ellipsometry (VASE)[3–5]. These methods can determine the average orientation of the TDM of the emission or absorption of molecules, e.g., in terms of the anisotropy factor $a \equiv \langle \cos^2\theta \rangle$, where $\theta$ is the angle between the TDM and the normal to the film, or through other related parameters[3,4,6]. Thereby, the orientation of emitters in OLEDs is commonly classified in relation to three limiting cases; namely, perfectly horizontal ($a = 0$), isotropic ($a = 1/3$), or perfectly vertical ($a = 1$) orientations. Crucially, by definition, currently used orientation parameters always average the orientation of the emitter molecules in the film over the entire ensemble and only give the first moment of their orientation distribution. Therefore, two ensembles of molecules that differ in width, peak value, skewness, or shape can have the same anisotropy factor (Fig. 1a, Supplementary Fig. 1). Measurement of the orientation distributions could provide valuable insight into the factors that drive molecular orientation, but this is currently not possible experimentally. Computational methods can model intermolecular interactions during film formation, which can in turn yield predictions of the orientation of molecules in thin films[4,17–22]. However, these models can only consider a limited number of atoms, and experimental validations beyond ensemble-averaged measurements remain elusive. Therefore, the ability to experimentally obtain the orientation distributions of emitters could provide significantly more information on the range of host-emitter interactions that drive molecular alignment in OLED-relevant systems.

Single-molecule fluorescence microscopy (SMFM) methods are ideal for probing the photophysics of materials on a molecule-by-molecule basis and have contributed significantly to our understanding of biological systems, chemical reactions, and luminescent materials[23]. Under certain conditions and with appropriate modifications, these methods can extract information about the orientation of individual molecular TDMs[24–29]. For example, single-molecule defocused orientation and position imaging (DOPI) has been employed to map conformational changes in biomolecules, as well as the morphology of lipid membranes and glassy polymers[30–34]. Great strides have been made to adapt SMFM to the study of organic semiconductors, such as conjugated polymers, dendrimers, perovskite nanocrystals, and OLED emitters[35–44]. However, those studies are mostly limited to solution-processed and solution-based systems. SMFM studies of OLEDs fabricated by thermal evaporation under high vacuum remain challenging, despite the prominence of these devices in basic research and the display industry. Furthermore, to our knowledge, SMFM has not been used to determine the orientation of the TDM of individual emitter molecules in device-relevant conditions.

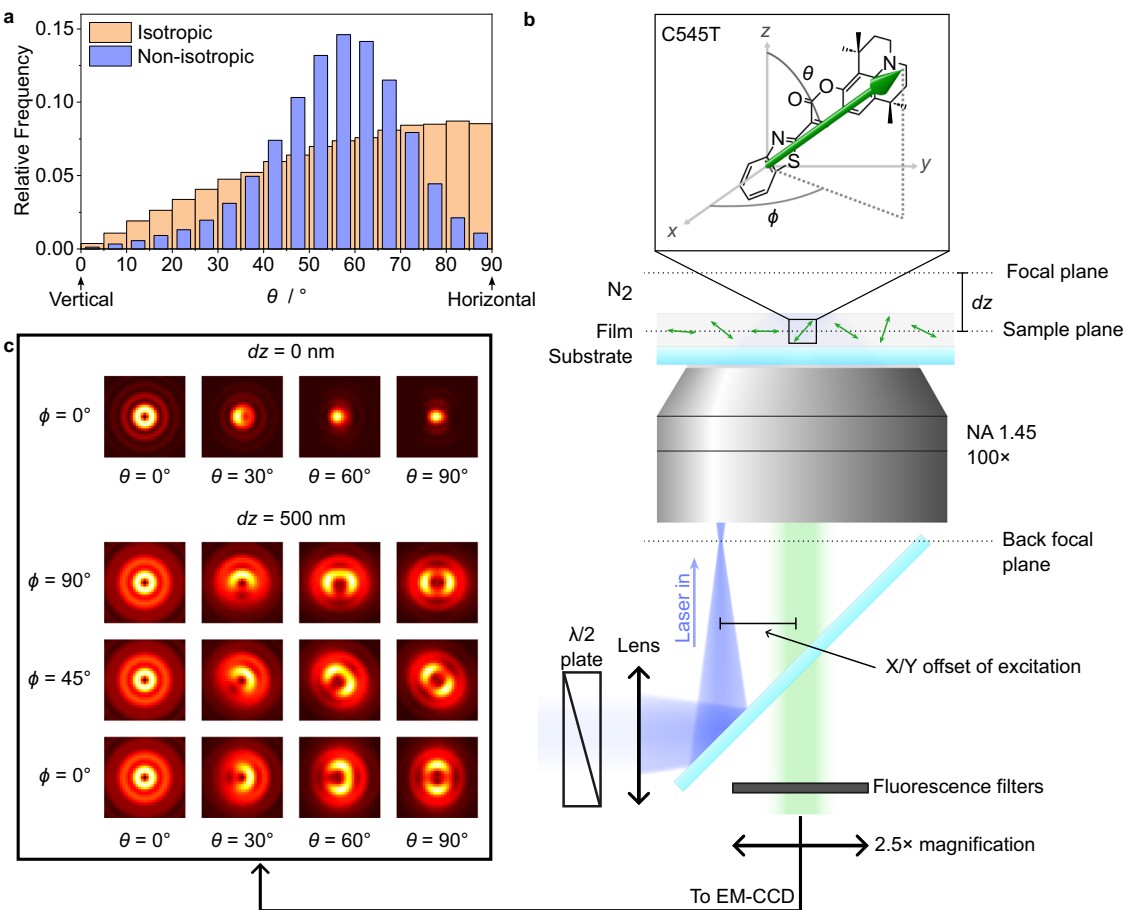

**Fig. 1 | Measurement of the orientation of single molecules via single-molecule defocused orientation and position imaging (DOPI). a** Illustrative comparison of an isotropic orientation distribution (proportional to $\sin\theta$) and a hypothetical non-isotropic distribution with a maximum at -57° and narrow width. The anisotropy factor is the same for both distributions ($a = 0.33$). The data for these hypothetical examples were obtained using a random-number generator (see Supplementary Methods). **b** Schematic of the microscope setup for single-molecule DOPI. A linearly polarized blue laser beam is focused on the back focal plane of a high-NA objective with a variable X/Y offset from the optical axis. This results in wide-field, variable-angle excitation of a film containing dispersed emitter molecules. The film is placed at $dz$ - 500 nm away from the focal plane of the objective to create intentionally defocussed images of the photoluminescence of individual emitter molecules on the EM-CCD camera. The inset shows the chemical structure of the hyperfluorescence-terminal emitter C545T and the direction of its $S_1 \rightarrow S_0$ transition dipole moment (TDM). Data taken from ref. 52. **c** Optical simulations of in-focus (top) and out-of-focus (bottom) single-molecule orientation patterns.

Here, we adapt and develop DOPI to measure the complete orientation distribution of the TDM of emitter molecules in state-of-the-art vacuum-deposited OLEDs. To achieve this, we thermally evaporate organic emitters at low enough rates to measure the photoluminescence from individual molecules. Furthermore, we eliminate the excitation bias commonly present in SMFM via a complementary-polarisation excitation scheme. Finally, we show that by optimized, thorough purification of a state-of-the-art OLED host material and careful management of the excitation and imaging conditions, we can attain sufficient signal-to-noise and signal-to-background ratios to experimentally measure the orientation distribution of thousands of single molecules of the hyperfluorescence-terminal emitter Coumarin 545T (C545T)[45]. Comparisons of measurements in inert solution-processed polymer matrices and OLED-relevant vacuum-deposited systems reveal that the orientation distribution of the emitter molecules in a doped film strongly depends on the processing conditions, and that the shape of the distributions can differ significantly—even for distributions with similar anisotropy factors. Moreover, we observe substantial differences in the orientation distribution of emitters located in different planes within the same film, which is particularly relevant for studying how the nano-scale morphology of films influences emitter orientation. These findings demonstrate how our technique can expand the current understanding of the factors that drive emitter orientation in OLEDs by providing an experimental means to link molecular-scale interactions to macroscopic device properties.

## Results

### Minimising excitation bias in defocused imaging of single emitter molecules

Figure 1b illustrates the general strategy employed to measure the orientation of individual molecules. The photoluminescence of spatially separated molecules in a thin film is imaged with the film positioned ~500 nm out of the focal plane of the microscope objective. The resulting diffraction patterns observed on the camera are specific to the polar and azimuthal orientation of the TDM of each molecule (Fig. 1c). The polar and azimuthal angles of the TDM relative to the plane of the film ($\phi$, $\theta$), as well as the three-dimensional position of the molecule in the film ($x$, $y$, $z$), can be determined by comparing and fitting the measured patterns to optical simulations via a least-square error minimisation algorithm (see Methods and Supplementary Methods for details of experimental conditions and data analysis).

For initial tests and referencing, we spin-coated thin films containing C545T doped at ~$10^{-6}$ wt% into the transparent polymer poly(methyl methacrylate) (PMMA, Supplementary Fig. 6). These samples were first imaged in-focus to determine the optimum concentration for orientation measurements (Supplementary Fig. 7). The number of emissive molecules per field of view (40×40 μm²) increased linearly with the concentration of C545T in the spin-coated solution, demonstrating that single molecules are resolvable and isolated from each other.

Next, single-molecule orientation patterns were imaged out of focus. Köhler illumination is usually used in epifluorescence excitation, i.e., the excitation beam travels along the normal to the sample plane, which can lead to sampling biases of an ensemble of fluorescent molecules because the electric field oscillates parallel to the horizontal plane. This issue was previously addressed by splitting the excitation beam using custom-built prism arrays[46]; this configuration allows the sample to be illuminated by two beams providing simultaneous in- and out-of-plane excitation. While useful for measuring dynamic processes in single molecules[47], this arrangement can lead to interference effects between the beams[46]. The static nature of our sample and the high photostability of OLED emitter materials allowed us instead to sequentially excite the sample along the three orthogonal directions

by displacing and focusing the excitation beam close to the edge of the back focal plane aperture of the objective (Fig. 1b). This strategy results in different wide-field excitation configurations, each with linear polarisation approximately along one of the three orthogonal axes of the sample which, in turn, depend on the linear polarisation and propagation direction of the beam (Fig. 2a). Each configuration excites a different subset of emitter molecules in a given field of view (Fig. 2b). Consequently, the measured distribution of the polar and azimuthal orientation angles strongly depends on the polarisation of the excitation beam (Supplementary Fig. 8). In order to avoid double-counting molecules in different excitation conditions, molecules that coincided in position and orientation angles across different measurements are identified and discarded (Fig. 2b) before all datasets are incorporated into a final set.

A histogram of the polar angles of C545T molecules in PMMA obtained in this manner exhibits a broad distribution that falls abruptly for $\theta < 30°$ (Fig. 2c). The mean $a$ value obtained directly from computing $\langle\cos^2\theta\rangle$ of this dataset is $0.32 \pm 0.01$, which is close to the value seen for an isotropic distribution. ARPL measurements of a reference sample (1 wt% C545T in PMMA) were in good agreement with those obtained from the single-molecule distribution ($a = 0.33 \pm 0.01$, Supplementary Fig. 9). However, the distribution obtained from our DOPI measurements differs from an isotropic distribution. This is more clearly visible for $\theta$ between 30° and 50°, where the distribution is higher than the isotropic limit, and towards 0° (vertical axis), where the frequency counts fall below it. This deviation from the isotropic case is likely due to interactions between the polymer host, the emitter molecules, and the substrate during film formation (see below). We emphasise that the complementary-polarisation excitation eliminates any bias towards exciting horizontally oriented molecules.

### Orientation distributions at different heights across a film

The defocused orientation patterns are sensitive to the vertical position of the individual molecules within the film (Supplementary Fig. 10). This allowed us to obtain orientation distributions for the TDMs of molecules at four different positions: near the interface to the glass substrate ($z = 0$ nm), 10 nm and 20 nm away from the substrate, and close to the top surface of the film ($z = 30$ nm) (Fig. 2d). Different peaks are distinguishable at each $z$-position (Fig. 2e): a narrow peak at 40° dominates the orientation distribution of molecules near the glass substrate, followed by a second peak at 70°. The distribution shifts towards larger angles for $z = 10$ nm, with a main peak at 80° and second peak still visible at 50°. The distribution shifts even further to horizontal orientations at $z = 20$ nm, where the peak remains at 80° and no other peaks are resolvable. Finally, the distribution shifts slightly back to smaller angles at $z = 30$ nm, even though the maximum is at 90°. From these distributions, we can also compute anisotropy factors for each position. The anisotropy factor is highest near the glass substrate ($a = 0.41$), reflecting the preferentially vertical orientation of the emitter TDMs ($a > 1/3$). As $z$ increases, $a$ reaches a minimum at $z = 20$ nm ($a = 0.25$, preferentially horizontal orientation) and shifts back to a less horizontal value near the top surface of the film ($a = 0.27$).

Based on these changes in orientation distribution across the height of the PMMA film, we hypothesise that the tendency of emitter TDMs to align horizontally within the film and near its top surface is a consequence of the centrifugal force during the spin-coating process, the intrinsic geometric anisotropy of the PMMA host and, potentially, that of the C545T emitters; as solvent molecules evaporate, preferentially horizontal polymer chains trap emitter molecules within the film. By contrast, the preferentially vertical orientation of the emitters near the glass substrate is likely a result of the interaction of PMMA and C545T with the substrate surface, presumably due to either friction or poor solvation at the boundary[48].

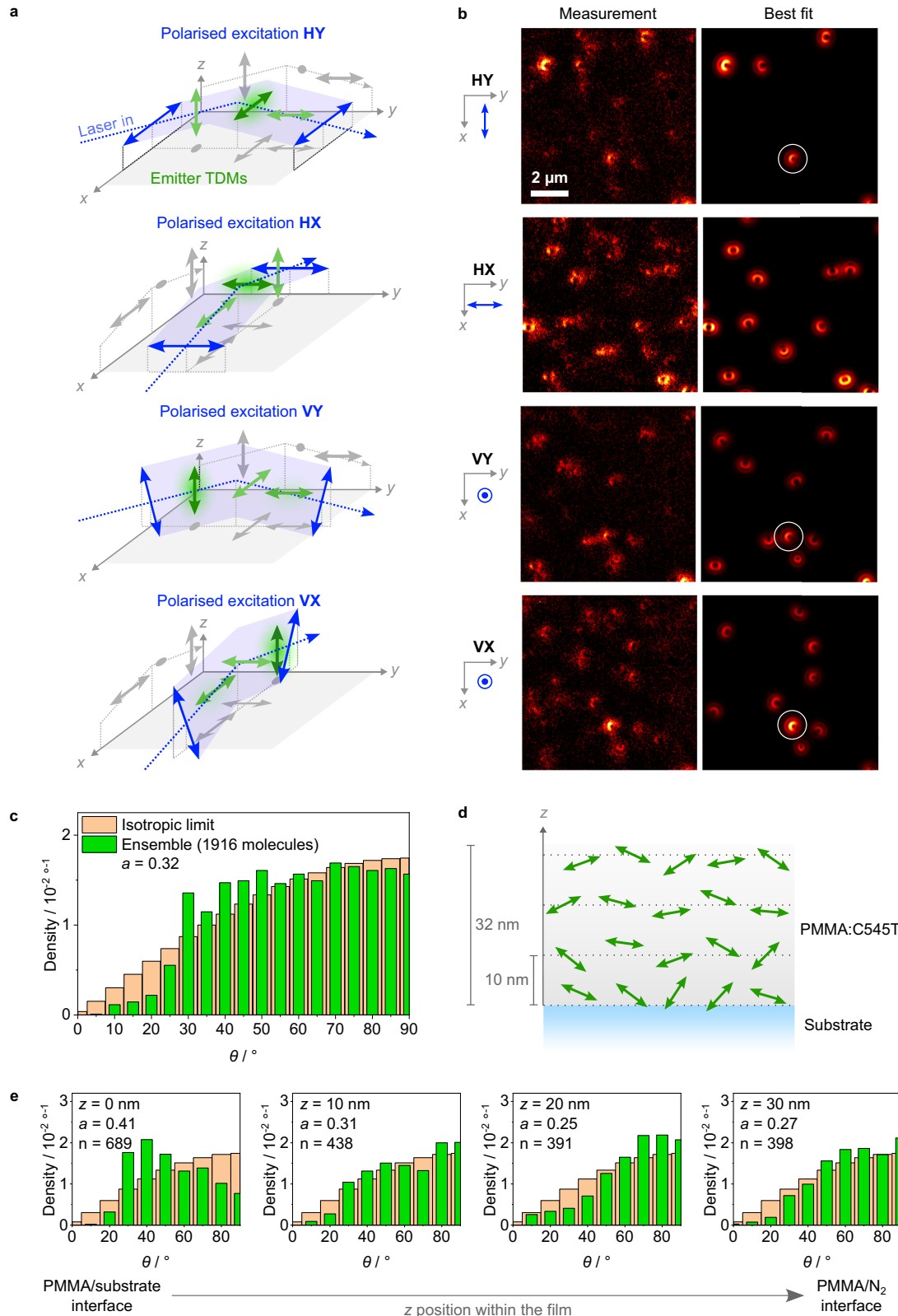

To summarize, even though the anisotropy factor measured for the ensemble case is close to that of the isotropic case, our single-molecule DOPI measurements revealed that the orientation of C545T molecules in PMMA is not isotropic across the film. Instead, it depends on the vertical position of the emitters within the film, and it is the averaging of the different distributions of the ensemble that leads to a value close to the isotropic limit in ARPL measurements.

## Single-molecule orientation distributions of emitters in OLED-relevant systems

Single-molecule imaging requires careful control over the number of target molecules in the field of view, as well as the numbers of fluorescent impurities that may compete with photoluminescence from the target molecules. To meet these requirements for vacuum-deposited films, evaporation rates far below the detection threshold of quartz

**Fig. 2 | Orientation distributions of C545T in PMMA. a** Illustration of the complementary-polarisation excitation conditions used for DOPI. Each configuration is characterized by the linear polarisation state of the excitation beam, which is parallel or perpendicular to the plane of the films (labelled H and V, respectively), and its direction of propagation along the XZ or YZ planes (labelled X and Y, respectively). **b** Examples of DOPI patterns measured with the excitation condition shown to the left of each panel. All measurements shown are for the same field of view. The best-fit optical simulations are shown for comparison. The patterns strongly depend on the polarisation of the laser beam. A representative pattern seen under HY, VX, and VY excitation is marked by a white circle. For visualisation

purposes only, an anisotropic background (coarse Gaussian filter) was subtracted from the experimental data to improve contrast. **c** Orientation distribution of polar angles for all detected C545T emitters. The distribution approaches the isotropic limit for large angles, but clearly deviates at smaller angles. **d** Schematic of a spin-coated film composed of C545T molecules dispersed in the transparent host PMMA. Horizontal dotted lines mark the different $z$ positions used for optical simulations. **e** Histogram of polar angles for emitters at different vertical positions ($z$) within the film. The orientation distributions are different for each $z$ value; this is also reflected in different anisotropy factors for each sub-ensemble.

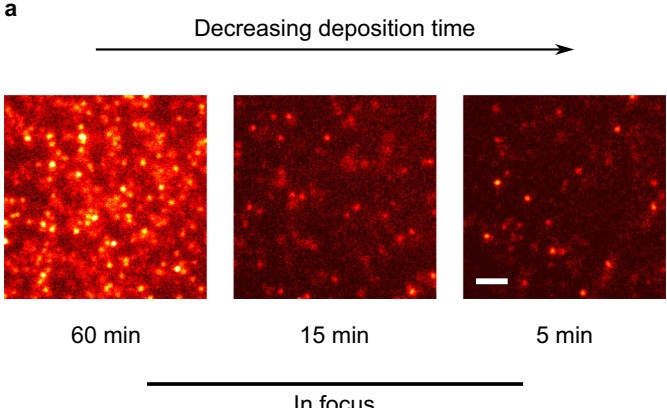

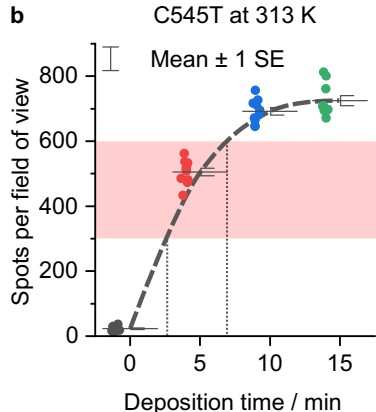

**Fig. 3 | Thermal evaporation of C545T molecules at concentrations suitable for SMFM. a** In-focus single molecule fluorescence images of vacuum-deposited C545T on glass substrates. Spots overlap for deposition times of 15 min and longer, but diffraction-limited spots from single molecule emission are resolvable when the deposition time is reduced to 5 min. Scale bar, 2 μm. For visualisation purposes only, a uniform background was subtracted to improve contrast. **b** Number of emissive spots per field of view (~ 40 × 40 μm²) as a function of C545T deposition

time. The number of resolvable spots saturates at long times, as emission from neighbouring molecules begins to overlap. The red shaded area indicates the target concentration range. The grey dashed line is shown as a guide to the eye. A deposition time of 0 min corresponds to substrates that were placed inside the chamber but remained covered during the whole process using the adjustable shutters.

crystal microbalance (QCM) sensors in commercial systems are needed, and photoluminescent impurities in both the deposition chamber and the host materials must be meticulously managed.

First, we deposited low-enough concentrations of C545T molecules on bare glass substrates by significantly decreasing the temperature of the evaporation crucible and carefully adjusting the duration of time that the substrates are exposed to incoming emitter molecules. For C545T, a crucible temperature of 313 K yielded a low-enough evaporation rate to enable control of the emitter concentration on the substrate by varying the deposition time with adjustable shutters (see Methods for further details). Depositions lasting a few minutes were sufficient to reliably disperse single emitter molecules on the substrate surface with enough separation to perform DOPI, i.e., ~500 molecules in a 40 × 40 μm² field of view (Fig. 3a, b). Importantly, no molecules were detected on substrates exposed to an empty crucible heated to 313 K for similar durations or when the substrate was exposed while the filled crucible was not actively heated (Supplementary Fig. 11), which confirmed that the measured molecules are the target C545T emitter.

In addition, we found that it is crucial to minimise the exposure of the substrates to fluorescent impurities from the evaporation chamber. Significant numbers of fluorescent molecules were found on substrates left in the evaporation chamber under high vacuum for long periods of time (> 3 h), even when no C545T was placed in the crucible (Supplementary Fig. 11). Presumably, this contamination results from desorption of fluorescent molecules from the walls or other parts of the evaporation chamber. Shielding the substrates at all times, except during the deposition process, effectively prevented contamination of the substrate with non-target molecules.

With a process for vacuum deposition of individual C545T molecules onto bare glass substrates established, we next measured the orientation distribution for these samples (Fig. 4a). Compared to C545T in PMMA, the molecules exhibited a significantly decreased brightness and photostability on bare glass, which led to a substantially reduced SNR of the defocused images (Fig. 4b). Given that all samples were encapsulated under nitrogen, we discard oxygen-driven irreversible photodegradation (photobleaching). However, the reduction in brightness of the emitter can also be due to the lower refractive index of the surrounding medium and/or fast, reversible quenching of the luminescence (blinking). The latter can be triggered by conformational changes—allowed here by the absence of a trapping matrix—and by the presence of photoluminescence quenching sites on the substrate[49,50], which can also lead to higher photobleaching rates[51]. We were still able to perform DOPI of these samples; however, only when using in-plane excitation with circularly polarised light. As expected for a relatively flat molecule like C545T[3,52], the resulting distribution of polar angles showed preferentially horizontal alignment of the TDMs of C545T molecules ($a = 0.12 \pm 0.03$), even after weighing the distribution with the different excitation probabilities inherent to in-plane excitation (Fig. 4c). The presence of orientation angles that deviate from purely horizontal may be a consequence of substrate roughness (Supplementary Fig. 12).

Next, we investigated the orientation distribution of C545T when co-deposited with the common OLED host 1,3-bis(N-carbazolyl)benzene (mCP; Supplementary Fig. 6). We sequentially deposited a 20 nm-thick spacer layer of pure mCP, a central 5 nm-thick co-deposited emissive layer, and a final 20 nm-thick capping layer of pure mCP (Fig. 4d). The spacer and capping layers ensure that the orientation of

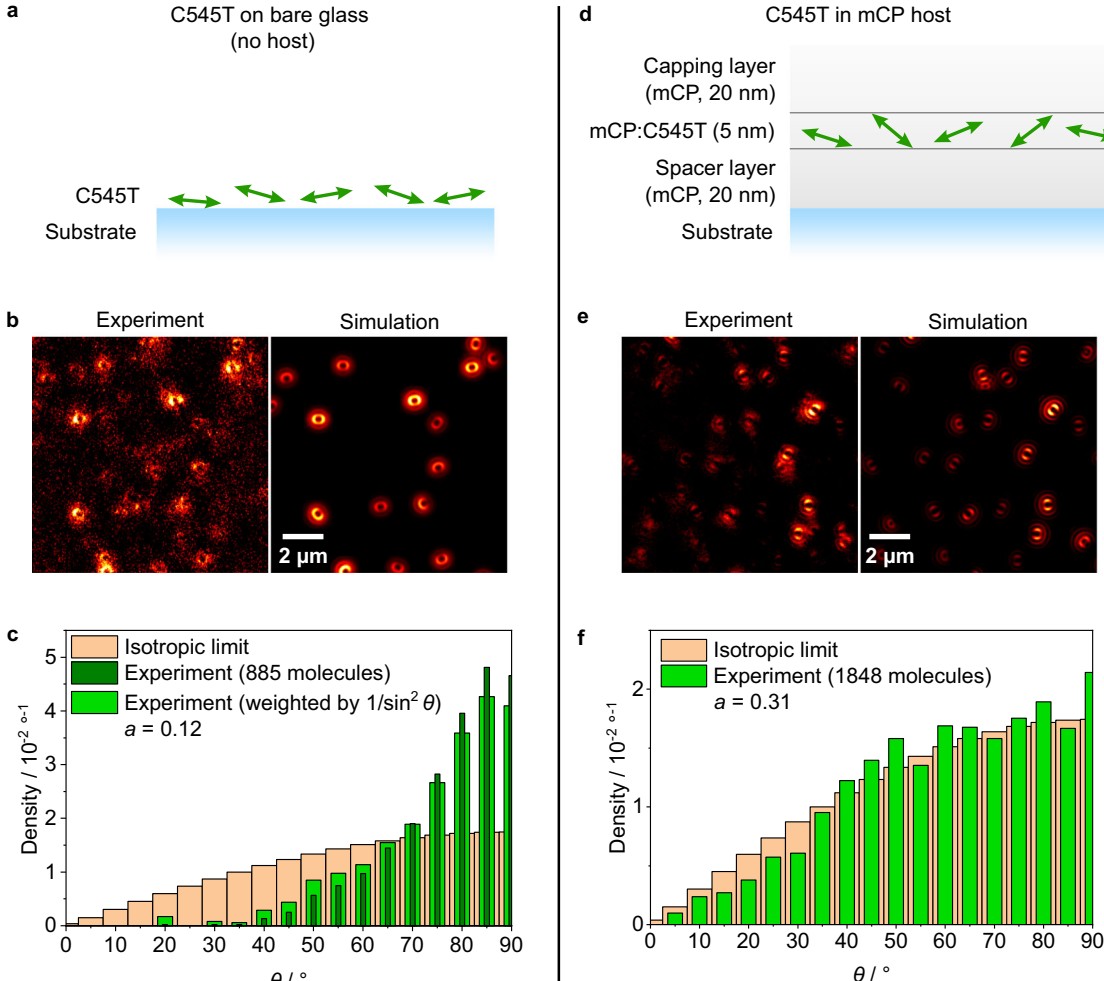

**Fig. 4 | Orientation distributions of C545T in vacuum-processed systems.**
**a** Illustration of TDMs of individual C545T molecules adsorbed on a glass substrate.
**b** Defocused orientation patterns measured with in-plane excitation with circularly polarised light. The best-fit optical simulations are shown for comparison.
**c** Distribution of polar angles of the TDMs of C545T emitters on glass, as measured (dark green) and after dividing by $\sin^2 \theta$ (with an artificial zero at $\theta = 0°$) to account for the excitation bias from the excitation configuration used in this measurement

(light green). **d** Illustration of the TDMs of individual C545T molecules dispersed in an mCP film, as fabricated in our experiments. **e** Representative defocused orientation patterns of C545T molecules in mCP using excitation configuration HY (see Fig. 2a). The best-fit optical simulations are shown for comparison. **f** Distribution of the polar angles of the TDMs of C545T emitters in mCP. An anisotropic background (coarse Gaussian filter) was subtracted from the experimental data in **b** and **e** to improve contrast for visualisation purposes only.

the emitter molecules is only driven by their interaction with the host, i.e., to minimise the influence of the substrate on the orientation of the emitter[53], and to enable host molecules in the capping layer to interact with buried emitter molecules, which might otherwise continue to diffuse on the surface of the film.

mCP shows very low absorption at the laser wavelength used for excitation (445 nm, Supplementary Fig. 13). To avoid background signal from any fluorescent impurities in the host material, we extensively purified mCP by three-fold vacuum sublimation. Combined with the high photostability and brightness of C545T, this provided a good SNR in the defocused patterns (Fig. 4e) and thus enabled reliable measurements of the TDM orientation distributions of C545T in mCP using our four configurations of complementary-polarisation excitation. The distribution of polar angles follows the isotropic limit more closely than in solution-processed PMMA films, albeit not exactly (Fig. 4f). The anisotropy factor computed from this distribution ($a = 0.31 \pm 0.01$), agrees well with ARPL measurements of the same emitter doped at 2 wt% in the same host ($a = 0.30 \pm 0.01$, Supplementary Fig. 14).

The near-isotropic distribution of C545T molecules is likely due to the relatively low molecular weight of C545T (430.5 g/mol) and the low glass transition temperature of mCP ($T_g \sim 338$ K)[54]. Due to increased

surface diffusion during deposition, small molecules are known to adopt orientations close to the isotropic limit when doped into materials with a low $T_g$, at least when considering the ensemble average of the orientation distribution[3,7,8,54]. Our measurements now confirm that the orientation is indeed randomised at the level of individual molecules.

## Discussion

By measuring the TDM orientation of individual molecules, we obtained complete orientation distributions of the OLED emitter C545T under a range of different conditions, including in a system relevant to vacuum-deposited hyperfluorescent OLEDs. To facilitate these measurements, we first established an excitation configuration that eliminates excitation bias. We then identified specific thermal evaporation conditions to deposit emitter molecules onto the substrate with sufficient separation to perform single-molecule DOPI. Combined with the use of a high-purity host, this approach enabled measurement of the TDM orientation of thousands of individual molecules in a high-throughput fashion. We also demonstrated the capability of this technique to unravel the different TDM orientation distributions of solution-processed samples with anisotropy factors

close to the isotropic limit. Remarkably, the orientation distribution of C545T in PMMA varied significantly at different vertical positions within the same film, which highlights the importance of accounting for possible inhomogeneities in TDM orientation in future orientation studies of both solution-processed and vacuum-deposited OLEDs.

The findings of this initial study illustrate how our technique will provide unprecedented insight into the factors that drive emitter orientation in OLEDs, particularly the various preferred orientations of a single emitter in different hosts, as well as at different positions within the same host layer. These results underline the importance of orientation distributions in our understanding of the factors that drive emitter orientation in OLEDs. While widely used measurement techniques such as VASE and ARPL provide accurate average values, emitter orientation cannot be reduced to these averages from a materials science perspective.

SMFM measurements are limited by the brightness and photostability of the emitter that is studied. Therefore, these measurements require molecules with relatively high radiative rates, which makes it challenging to apply our technique to phosphorescent compounds. However, fluorescent and state-of-the-art TADF emitters are good candidates for further studies that use DOPI to map the complete orientation distribution of an emitter.

In the future, the combination of DOPI measurements of vacuum-processed samples with quantum-chemical calculations of the same systems (e.g., molecular dynamics or Monte Carlo simulations) might contribute significantly to our understanding of how these preferred orientations occur and shed light on the importance of different intermolecular interactions for the orientation of OLED emitters. The low concentrations of emitters in samples used for single-molecule microscopy will make it possible to study the influence of host-emitter interactions on emitter orientation in the absence of artifacts originating from emitter-emitter interaction. Importantly, the comparison of these results with ensemble-averaged measurements of samples with emitter concentrations in the 1–10 wt% range can then also bring further clarification of the relevance of emitter-emitter interactions on their molecular orientation and photophysics. Moreover, the high level of control on sample architecture offered by the use of vacuum deposition and the spatial resolution provided by DOPI will help to address open questions such as the influence of interfaces—between the substrate and organic molecules, and also between different layers within an OLED—and the degree of order of the host on the orientation of emitter molecules.

## Methods

### Materials
C545T was obtained from Lumtec and used without further purification. mCP was obtained from Sigma Aldrich and purified in-house via three-fold vacuum sublimation using a Creaphys DSU05. PMMA was obtained from Sigma Aldrich and purified by three subsequent cycles of dilution in toluene and precipitation in isopropanol. KOH, acetone, isopropanol, methanol, and toluene were obtained from Fisher Scientific; all solvents were HPLC-grade or higher.

### Substrate cleaning
For SMFM, borosilicate glass substrates (No. 1.5) were cleaned using the following cycle: 15 min ultrasonication in acetone, 15 min ultrasonication in methanol, rinse with milli-Q water, 5 min ultrasonication in milli-Q water, 15 min ultrasonication in 1 M KOH in milli-Q water, rinse twice with milli-Q water, 5 min ultrasonication in isopropanol (twice), drying under nitrogen flow, and 10 min treatment in an oxygen plasma system (for solution-processed samples) or in a UV-ozone cleaner (for vacuum-processed samples). Samples were prepared shortly after substrate cleaning to avoid contamination. For ARPL measurements, EagleXG glass substrates were cleaned by 15 min ultrasonication in acetone, 15 min ultrasonication in isopropanol, and finally 3 min oxygen-plasma treatment. For VASE measurements, silicon substrates with a native silicon oxide layer were cleaned in the same way as the glass substrates for ARPL, but the oxygen-plasma treatment was omitted to avoid thickening of the silicon oxide layer.

### Fabrication of solution-processed samples for SMFM
C545T was diluted to ~5 nM in toluene via serial dilutions. For concentrations diluted to 100 nM or lower, only glassware (vials and pipettes) cleaned in the same way as the substrates were allowed to come into contact with the solutions. PMMA was diluted to ~30 mg/mL. Final dilutions of PMMA at ~10 mg/mL plus C545T at 0, ~100, ..., and ~500 pM were prepared in toluene and gently shaken for 15 min, then spin-coated at 2000 RPM for 30 s. Finally, the films were annealed at 100 °C for 10 min inside a nitrogen-filled glovebox to evaporate residual solvent, encapsulated with a cavity glass lid, and stored under nitrogen until the SMFM measurements were performed.

### Fabrication of vacuum-processed samples for SMFM
Thermal evaporation was carried out in a commercial high-vacuum deposition system (Angstrom EvoVac) equipped with actively temperature-controlled furnaces for evaporation of organic materials (Luxel Radak I). Once placed inside the vacuum chamber, the substrates were protected by adjustable shutters placed ~2 mm away from their surface at all times, other than during the deposition step. All depositions were carried out at chamber pressures $<5 \times 10^{-7}$ Torr. The deposition rate of C545T was controlled by keeping the crucible in the furnace at a pre-determined constant temperature. The deposition rate of the mCP host was set to 0.3 Å/s and controlled through a PID feedback loop using deposition rate readings from QCM sensors. All samples were encapsulated with a cavity glass lid inside a nitrogen-filled glove box without exposure to ambient conditions. All measurements were conducted shortly after fabrication to avoid recrystallisation of the host.

### Fabrication of samples for ARPL
For solution-processed samples, a toluene solution of PMMA (~10 mg/mL) and C545T (~0.1 mg/mL) was spin-coated onto either EagleXG glass or silicon substrates at 2000 RPM for 30 s. The films were annealed at 100 °C for 10 min inside a nitrogen-filled glovebox to evaporate residual solvent. Films for ARPL were encapsulated with a cavity glass lid under nitrogen, and all samples were stored under nitrogen until measured.

For vacuum-deposited samples, 40 nm-thick films of C545T doped at 2 wt% in mCP were deposited at 0.6 Å/s on EagleXG glass substrates and on silicon substrates using the thermal-evaporation system described above. Films on glass substrates were immediately encapsulated with a cavity glass lid in a nitrogen-filled glove box. All samples were measured within 24 h to avoid recrystallisation of the host material.

### Film characterisation via VASE
The thickness and optical constants of neat PMMA and mCP films, as well as of the films for ARPL measurements, were quantified via VASE (J.A. Woollam M2000 and subsequent modelling in software CompleteEASE). These data (Supplementary Figs. 15 and 16) were used for analysis of the DOPI and ARPL data.

### Single-molecule fluorescence microscopy
SMFM measurements were carried out on a Nikon Eclipse Ti-2 microscope equipped with a Perfect Focus System. Samples were excited with a Vortran Stardus CW laser (445 nm). The linear polarisation of the beam was controlled using a half-wave plate. The beam shape was cleaned and expanded using a telescope array and pinhole and cropped by a movable iris to produce an off-axis beam focused on the back-focal plane of an Olympus oil-

immersion objective (NA 1.45, 100×). The power density of the laser at the sample plane was ~36 W/cm$^2$. The light collected by the objective was passed through a dichroic (Semrock Di02-R442-25×36), and then through two further fluorescence filters (Semrock FF01-540/50-25 and Thorlabs FBH520-10) to discard fluorescence from non-target molecules. Finally, the image was expanded by a further 2.5× magnification and recorded using a Peltier-cooled EMCCD (Andor iXon Ultra 888), giving a field of view of ~40 × 40 μm$^2$. The samples were first imaged in focus to quantify the number of molecules per field of view (Fig. 3, Supplementary Figs. 7 and 17). Then, selected samples were imaged out-of-focus by displacing the objective to a fixed distance from the substrate using the Perfect Focus System.

### Image processing
All image-processing and optical simulations were performed using MATLAB software (The MathWorks, Inc.). In-focus images to quantify the number of spots per field of view were processed using peak-finding and drift-correction algorithms obtained from the single-molecule image analysis package iSMS[55]. Optical simulations were obtained from the QDControl package from Enderlein et al. [30,56]. The pattern-matching algorithm of our analysis of defocused images followed the process reported by Patra et al. [31] but was modified by the addition of a segmentation algorithm to decrease the number of false positives. Additional algorithms for identifying double counts between different excitation configurations were developed in-house. The complete image processing workflow is available in ref. [57].

### Angle-resolved photoluminescence spectroscopy measurements
ARPL of films deposited on EagleXG glass substrates was performed using a previously described custom-built setup[58]. The data were fitted to optical simulations using the anisotropy factor of the average transition dipole of the emitters and the film thickness as free parameters[59].

### Atomic-force microscopy measurements
All atomic-force microscopy (AFM) measurements were performed with an Asylum Research AFM (Oxford Instruments, MFP − 3D Infinity) using alternating contact mode. For all measurements AC200TS cantilevers (Olympus) were used. Data evaluation was performed using the software Gwyddion. The mean square roughness of all samples was estimated using the statistical analysis feature of Gwyddion. Beforehand, a plane level subtraction and a scan line artefact removal were performed.

## Data availability
The research data supporting this publication can be accessed at https://doi.org/10.17630/c835efe3-402d-4a9d-8ea0-afc5bad6efd2[57]. Source data are provided in this paper.

## Code availability
The custom software used to analyse single-molecule fluorescence microscopy images is available at https://doi.org/10.17630/c835efe3-402d-4a9d-8ea0-afc5bad6efd2[57].

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

## Acknowledgements

We are grateful to Prof. J. Carlos Penedo for his advice and technical support on single-molecule microscopy and for providing the related laboratory facilities at the early stages of this project. This work was supported by the Volkswagen Foundation (No. 93404) and the DFG-funded Research Training Group "Template-Designed Organic Electronics (TIDE)", RTG2591. M.C.G. acknowledges support from the Alexander von Humboldt Stiftung through the Humboldt-Professorship. A.M. acknowledges funding from the European Union's Horizon 2020 research and innovation programme under Marie Skłodowska-Curie grant agreement No. 101023743 (PolDev).

## Author contributions

F.T.C. and M.C.G. designed the study and planned the experiments. F.T.C. fabricated the solution-processed samples, performed the SMFM measurements, wrote the custom-built SMFM data-processing algorithms, and performed ARPL measurements of the reference samples. D.H. purified the host material for vacuum-processed samples. S.H. and F.T.C. designed and fabricated the thermally evaporated samples. A.M. and F.T.C. performed the ellipsometry measurements and related analysis. A.G. performed preliminary SMFM studies of solution-processed samples and assisted in the analysis of DOPI data. P.W. performed the AFM measurements of the surface of the glass substrates under supervision of K.M. All authors analysed and interpreted the results. F.T.C. and M.C.G. wrote the manuscript with contributions from all authors.

## Funding

## Competing interests

The authors declare no competing interests.
