## [Peer Review File · Nature Communications]

Orientation distributions of vacuum-deposited organic emitters revealed by single-molecule microscopyREVIEWER COMMENTS

Reviewer #1 (Remarks to the Author):

The authors use defocused single-molecule fluorescence imaging to measure 3D orientation of small dye molecules embedded in evaporated films of a charge-transporting OLED host matrix. They succeeded in preparing samples at single-molecule level concentration within the 45 nm thick films with sufficiently low fluorescence background, which itself is an enormous task. They then used the DOPI technique to find out that the molecules are oriented almost isotropically.

Overall, the use of the DOPI technique to characterize OLED materials is original, and the successful demonstration will surely have impact not only in the OLED field but in molecular photophysics in general. I would recommend publication after the authors address the points below:

1. The Introduction makes an impression that it is missing a part because the references 23-40 are not mentioned in the text and DOPI appears without any explanation of its meaning.
2. Several original and innovative techniques (some of them similar to the one in the current submission) for the correct measurement of 3D orientation of single molecules were proposed and demonstrated in the earlier stages of single molecule research and it would be fair to acknowledge these works (e.g., J. Chem. Phys. 118 (2003) 9824; J. Chem. Phys. 118 (2003) 5279; J. Opt. Soc. Am. B 21 (2004) 1210; Eur. Phys. J. D 28 (2004) 67; Appl. Phys. Lett. 86 (2005) 121104).
3. I would disagree with the statement on page 8, film. 'It is commonly assumed that small molecules adopt an isotropic orientation during the spin-coating process, but our single molecule-resolved data show that this is not the case here.' On contrary, I think the common assumption would be that small molecules get oriented during spin-coating in polymer matrix due to the shear forces present in the process. In any case, orientation of PDI molecules during spin-coating in PMMA was demonstrated earlier (Phys. Chem. Chem. Phys. 13 (2011) 6970).
4. It is interesting to observe the position-dependent orientation of the molecules along the PMMA film thickness in Fig. 2d, e. What is the error in the z-position determination from the defocused images? Please discuss.
5. The authors found that evaporation of the molecules on bare glass results in preferentially horizontal orientation of the transition dipoles (Fig. 4c). However, in the experimental and simulated defocused images of this sample in Fig. 4b the symmetrical defocused patterns indicate more-or-less vertical orientation of the molecules (also in comparison with Fig. 4e). Please explain. Also, the deviation from complete horizontal orientation is explained by glass surface roughness – have you measured that? Is that reason plausible?
6. The lower brightness of the molecules on bare glass is explained due to residual charges from the cleaning process. This sounds speculative – what step of the cleaning process would lead to the charging? Do you actually observe blinking on the bare glass, is it stronger than in PMMA? Isn't the

higher brightness in PMMA due to the higher refractive index, and/or due to the rigid environment of the polymer preventing conformational changes of the dye?

7. Minor point: the schematic depiction of the molecular orientation in Fig. 4a and 4d is same but you probably want to make the point that these are different. I suggest to reflect this in the schemes.

Reviewer #2 (Remarks to the Author):

The manuscript presents an advancement in the field of organic light-emitting diodes (OLEDs). The authors address a significant knowledge gap by using a cutting-edge measurement technique called single-molecule defocused orientation and position imaging (DOPI). This innovative approach enables the measurement of transition dipole orientation at the individual molecule level, providing insights into emitter orientation and its impact on device performance. The authors employ a complementary polarisation excitation technique that effectively minimizes excitation bias, ensuring accurate and reliable measurements. Furthermore, they successfully achieved ultra-low doping concentrations of the hyperfluorescence-terminal OLED emitter C545T, resulting in a comprehensive analysis of orientation distributions for over a thousand individual emitter molecules. One of the significant achievements of this study is the exploration of the influence of the emitter's environment on its orientation distributions. The researchers analyze both solution-processed and vacuum-processed systems, unveiling the striking dependence of orientation distributions on the surrounding environment. This observation opens up new avenues for investigating and manipulating emitter orientation in various material systems for OLEDs, providing valuable knowledge for the development of future OLED technologies.

The work described in this manuscript holds immense promise for the OLED research community. By overcoming the limitations of previous measurement techniques that only provided mean orientation values, the authors pave the way for attaining unprecedented information on the factors that determine emitter orientation. This will undoubtedly catalyze advancements in the design and fabrication of OLED devices, leading to improved device performance, efficiency, and overall user experience.

The manuscript is written very clearly, the results are convincing, and the topic is of significant importance. I could not find any flaw in the presented material, and I recommend publication as is.

Reviewer #3 (Remarks to the Author):

Exploring and manipulating the anisotropic distributions of the transition dipole moment of emitter molecules are particularly important for the performance of organic light-emitting diodes (OLEDs). In this work, the authors declared that they adapted and developed single-molecule defocused orientation and position imaging to measure the transition dipole orientation of individual molecules. By analyzing a solution-processed system and two vacuum-processed systems, they revealed that the orientation distributions strongly depend on the environment of the emitter. Overall, the research is somewhat interesting and can supply some information on the factors of the emitter orientation. However, the proposed method has been widely used for single-molecule spectroscopy, thus showing indistinctively scientific significance. Furthermore, some experimental results are inscrutable. My comments are as follows:

1. The authors declared that the innovation of this work is the adaption and development of single-molecule defocused orientation and position imaging. However, the defocused single-molecule imaging was developed many years ago, and has been applied to investigate the orientation of organic molecules, quantum dots, and other nanomaterials, such as publications [Adv. Mater., 2009, 21, 1079; J. Phys. Chem. B, 2012, 116, 12878; ACS Nano, 2012, 6, 1268; J. Phys. Chem. Lett., 2014, 5, 2830; et al.,]. Compared with these works, what's the scientific significance of this work? What are the major problems solved by this work? Furthermore, the motivation of this work is to explore the orientation of the emitters in OLEDs. However, the emitters are pretty dense for OLEDs, far from the applied requirement of single-molecule technology. How do the authors plan to address this obstacle?
2. What is the spatial resolution of the Z-axis of this system? Figure 2e presents the histogram of polar angles for emitters at different vertical positions within the film. This result is incomprehensible.
3. The main conclusion of this work is that "the orientation distributions strongly depend on the environment of the emitter." I think this conclusion is obvious. Can this work supply any further deep insights?

Reviewer #1 (Remarks to the Author):

The authors use defocused single-molecule fluorescence imaging to measure 3D orientation of small dye molecules embedded in evaporated films of a charge-transporting OLED host matrix. They succeeded in preparing samples at single-molecule level concentration within the 45 nm thick films with sufficiently low fluorescence background, which itself is an enormous task. They then used the DOPI technique to find out that the molecules are oriented almost isotropically. Overall, the use of the DOPI technique to characterize OLED materials is original, and the successful demonstration will surely have impact not only in the OLED field but in molecular photophysics in general. I would recommend publication after the authors address the points below:

We thank the reviewer for their encouraging feedback. All points raised have been addressed in the revised version of our manuscript as detailed below.

1. The Introduction makes an impression that it is missing a part because the references 23-40 are not mentioned in the text and DOPI appears without any explanation of its meaning.

We thank the reviewer for alerting us to this issue and sincerely apologize for this oversight. Indeed, it appears that we have lost the paragraph introducing single-molecule microscopy and defining the acronym for defocused orientation and position imaging (DOPI). This has now been re-introduced as the first paragraph of page 4 of our revised manuscript. In response to several of the points brought up by the reviewers, we have also expanded and thoroughly revised this paragraph; it now reads as follows:

Single-molecule fluorescence microscopy (SMFM) methods are ideal for probing the photophysics of materials on a molecule-by-molecule basis and have contributed significantly to our understanding of biological systems, chemical reactions, and luminescent materials.²⁴ Under certain conditions and with appropriate modifications, these methods can extract information about the orientation of individual molecular TDMs.^{25–30} For example, single-molecule defocused orientation and position imaging (DOPI) has been employed to map conformational changes in biomolecules, as well as the morphology of lipid membranes and glassy polymers.^{31–35} Great strides have been made to adapt SMFM to the study of organic semiconductors, such as conjugated polymers, dendrimers, perovskite nanocrystals, and OLED emitters.^{36–45} However, those studies are mostly limited to solution-processed and solution-based systems. SMFM studies of OLEDs fabricated by thermal evaporation under high vacuum remain challenging, despite the prominence of these devices in basic research and the display industry. Furthermore, to our knowledge, SMFM has not been used to determine the orientation of the TDM of individual emitter molecules in device-relevant conditions.

We have also verified all references and double-checked that all acronyms are introduced on their first appearance in the text.

2. Several original and innovative techniques (some of them similar to the one in the current submission) for the correct measurement of 3D orientation of single molecules were proposed and demonstrated in the earlier stages of single molecule research and it would be fair to acknowledge these works (e.g., J. Chem. Phys. 118 (2003) 9824; J. Chem. Phys. 118 (2003) 5279; J. Opt. Soc. Am. B 21 (2004) 1210; Eur. Phys. J. D 28 (2004) 67; Appl. Phys. Lett. 86 (2005) 121104).

We thank the reviewer for pointing this out. We have added these references (26-30 in our revised manuscript) and now acknowledge these earlier developments of single-molecule orientation imaging techniques in the first paragraph of page 4 mentioned in our response to the previous point. The relevant text reads:

Under certain conditions and with appropriate modifications, these methods can extract information about the orientation of individual molecular TDMs.^{25–30}

3. I would disagree with the statement on page 8, film. ‘It is commonly assumed that small molecules adopt an isotropic orientation during the spin-coating process, but our single molecule-resolved data show that this is not the case here.’ On contrary, I think the common assumption would be that small molecules get oriented during spin-coating in polymer matrix due to the shear forces present in the process. In any case, orientation of PDI molecules during spin-coating in PMMA was demonstrated earlier (Phys. Chem. Chem. Phys. 13 (2011) 6970).

We thank the reviewer for their comment and for pointing us to this reference. Our claim comes from a common assumption in the OLED community that solution-processed small molecules have an isotropic orientation (see, e.g., Phys. Rev. Applied 8, 037001 (2017)). However, we agree that this is not as clear cut as was stated in our original manuscript and, since this is also not the central point of our work, we have removed the aforementioned statement in the revised manuscript. The paragraph (page 9, second-to-last paragraph) now reads:

To summarize, even though the anisotropy factor measured for the ensemble case is close to that of the isotropic case, our single-molecule DOPI measurements revealed that the orientation of C545T molecules in PMMA is not isotropic across the film. Instead, it depends on the vertical position of the emitters within the film, and it is the averaging of the different distributions of the ensemble that leads to a value close to the isotropic limit in ARPL measurements.

4. It is interesting to observe the position-dependent orientation of the molecules along the PMMA film thickness in Fig. 2d, e. What is the error in the z-position determination from the defocused images? Please discuss.

We thank the reviewer for this question. We have estimated the error in the z-position of the molecules from the inverse-square error of each fit as:

$$\Delta z = z^{(r)} - z^{(r+l)} + \Delta z_{min},$$

where Δz_{min} is the error related to having a minimum step size in the pattern library, and l indicates the l^{th} -nearest neighbour to the best-fitted pattern $p^{(r)}$ in the library of patterns such that:

$$e_{mn}^{(r)} = \sum_{j,k} s_{jk} \left(x_{m+j,n+k} - c_{mn}^{(r)} p_{jk}^{(r)} - d_{mn}^{(r)} b_{jk} \right)^2 < c_{mn}^{(r)} \sum_{j,k} s_{jk} \left(p_{jk}^{(r)} - p_{jk}^{(r+l)} \right)^2.$$

Here, x denotes the measured pattern, c is the fitted intensity of the molecule, d is the fitted background at the position of the molecule position (j,k) , and e is the error in the fit. In this way, the estimated error in the z-position of the molecules satisfies the condition that the squared difference between the patterns $p^{(r)}$ and $p^{(r+l)}$ is larger than the squared difference between the best-fitted pattern and the experimental data.

To further clarify this point, we have added the following description of our error analysis to pages 2 and 3 of the revised Supplementary Information, in which we discuss our estimation of errors in general terms for the free-fitting parameters $\xi = \vartheta, \varphi, z$ as follows:

In order to fit the z-position of each molecule, we first needed to determine the in-focus position of the sample (i.e., $dz = 0$). This was achieved by comparing the patterns from the brightest molecules to optical simulations using a modified version of the software QDControl. The selected value was then fixed for all other identified patterns in the same dataset (525 nm for the solution-processed samples, and 500 nm for the experiments for the thermally-evaporated samples).

[...]

The error in each of the free-fitting parameters ($\xi = \vartheta, \varphi, z$) was estimated from the inverse-square error of each fit as

$$\Delta\xi = \xi^{(r)} - \xi^{(r+l)} + \Delta\xi_{min}$$

where $\Delta\xi_{min}$ is the error related to having a minimum step size for each parameter in the pattern library, and l indicates the l th-nearest neighbour to the best-fitted pattern $p^{(r)}$ in the library such that

$$e_{mn}^{(r)} = \sum_{j,k} s_{jk} \left(x_{m+j,n+k} - c_{mn}^{(r)} p_{jk}^{(r)} - d_{mn}^{(r)} b_{jk} \right)^2 < c_{mn}^{(r)} \sum_{j,k} s_{jk} \left(p_{jk}^{(r)} - p_{jk}^{(r+l)} \right)^2.$$

In this way, the estimated error in each parameter ξ satisfies the condition that the squared difference between the patterns $p^{(r)}$ and $p^{(r+l)}$ is larger than the squared difference between the best-fitted pattern and the experimental data. $\Delta\xi_{min}$ was set to the minimum step size of each parameter's grid unless $\xi^{(r+l)}$ coincided with the boundaries set for each parameter (0° and 90° for ϑ , and 0 nm and 30 nm for z in the experiments involving solution-processed samples), in which case $\Delta\xi_{min}$ was set to $\frac{1}{2}$ of the minimum grid step.

We have also added histograms of the distribution of error values and plots of such values against the weighted inverse error ($L = c/\sqrt{e}$) from all our experiments in Figures S3, S4, and S5.

5. The authors found that evaporation of the molecules on bare glass results in preferentially horizontal orientation of the transition dipoles (Fig. 4c). However, in the experimental and simulated defocused images of this sample in Fig. 4b the symmetrical defocused patterns indicate more-or-less vertical orientation of the molecules (also in comparison with Fig. 4e). Please explain. Also, the deviation from complete horizontal orientation is explained by glass surface roughness – have you measured that? Is that reason plausible?

We believe there is a misunderstanding here; the symmetrical defocused patterns in Figure 4b originate from a close-to-horizontal rather than preferentially vertical alignment of the molecules. In contrast to the circularly symmetrical patterns of vertically aligned molecules, the patterns in Fig. 4b mentioned by the reviewer are more elongated and have two lobes or an open, elongated “C” shape (see Figure 1c for comparison). We appreciate that the two lobes from horizontally aligned molecules are clearer in Figure 4e; this is due to the difference in the refractive indices of the surrounding media (air and mCP).

We have now performed atomic force microscopy of the bare glass (Figure S12 in our revised manuscript). These yielded an RMS roughness of 2.14 nm. This value is in the same order of magnitude as the size of the C545T emitter molecules, which is 1.55 nm. Therefore, we think that it is quite plausible that the roughness of the glass affects the orientation of the C545T molecules when evaporating the molecules on bare glass.

6. The lower brightness of the molecules on bare glass is explained due to residual charges from the cleaning process. This sounds speculative – what step of the cleaning process would lead to the charging? Do you actually observe blinking on the bare glass, is it stronger than in PMMA? Isn't the higher brightness in PMMA due to the higher refractive index, and/or due to the rigid environment of the polymer preventing conformational changes of the dye?

We thank the reviewer for their comment. We have not performed a detailed characterisation of blinking from emitter molecules. However, we do observe faster photobleaching for emitter molecules deposited on bare glass than for those in the PMMA matrix.

It is known that the UV/ozone treatment of the glass substrates yields a high density of reactive silanol groups on their surface (Biomicrofluidics 5, 036501 (2011)). Previous single-molecule studies with fluorescent emitters have attributed blinking and photobleaching to interaction with charges and with the surface of glass substrates (e.g., J. Phys. Chem. A 2003, 107, 35, 6770–6776 and J. Phys. Chem. Lett. 2011, 2, 21, 2827–2831). However, we agree with the reviewer that at this stage we cannot conclusively assign the lower brightness to residual charges. We also agree with the reviewer that the higher brightness of the emitters in PMMA might also be a result of the rigidity and higher refractive index of the local environment.

As knowing the exact origin of the observed difference in brightness is not crucial to our conclusions, we have addressed this point by incorporating the alternative explanations brought up by the reviewer into the first paragraph on page 12 of our revised manuscript, as well as by removing the speculative explanation involving the cleaning of the substrate. The paragraph now reads:

...Given that all samples were encapsulated under nitrogen, we discard oxygen-driven irreversible photodegradation (photobleaching). However, the reduction in brightness of the emitter can also be due to the lower refractive index of the surrounding medium and/or fast, reversible quenching of the luminescence (blinking). The latter can be triggered by conformational changes—allowed here by the absence of a trapping matrix—and by the presence of photoluminescence quenching sites on the substrate,^{50,51} which can also lead to higher photobleaching rates.⁵²...

7. Minor point: the schematic depiction of the molecular orientation in Fig. 4a and 4d is same but you probably want to make the point that these are different. I suggest to reflect this in the schemes.

We thank the reviewer for their suggestion. We have modified the diagrams in Figure 4a and 4d to better illustrate the difference between the orientation distribution of C545T deposited on bare glass and trapped in mCP.

Reviewer #2 (Remarks to the Author):

The manuscript presents an advancement in the field of organic light-emitting diodes (OLEDs). The authors address a significant knowledge gap by using a cutting-edge measurement technique called single-molecule defocused orientation and position imaging (DOPI). This innovative approach enables the measurement of transition dipole orientation at the individual molecule level, providing insights into emitter orientation and its impact on device performance. The authors employ a complementary polarisation excitation technique that effectively minimizes excitation bias, ensuring accurate and reliable measurements. Furthermore, they successfully achieved ultra-low doping concentrations of the hyperfluorescence-terminal OLED emitter C545T, resulting in a comprehensive analysis of orientation distributions for over a thousand individual emitter molecules. One of the significant achievements of this study is the exploration of the influence of the emitter's environment on its orientation distributions. The researchers analyze both solution-processed and vacuum-processed systems, unveiling the striking dependence of orientation distributions on the surrounding environment. This observation opens up new avenues for investigating and manipulating emitter orientation in various material systems for OLEDs, providing valuable knowledge for the development of future OLED technologies.

The work described in this manuscript holds immense promise for the OLED research community. By overcoming the limitations of previous measurement techniques that only provided mean orientation values, the authors pave the way for attaining unprecedented information on the factors that determine emitter orientation. This will undoubtedly catalyze advancements in the design and fabrication of OLED devices, leading to improved device performance, efficiency, and overall user experience.

The manuscript is written very clearly, the results are convincing, and the topic is of significant importance. I could not find any flaw in the presented material, and I recommend publication as is.

We thank the reviewer for their exceptionally positive assessment of our work and for their encouraging feedback.

Reviewer #3 (Remarks to the Author):

Exploring and manipulating the anisotropic distributions of the transition dipole moment of emitter molecules are particularly important for the performance of organic light-emitting diodes (OLEDs). In this work, the authors declared that they adapted and developed single-molecule defocused orientation and position imaging to measure the transition dipole orientation of individual molecules. By analyzing a solution-processed system and two vacuum-processed systems, they revealed that the orientation distributions strongly depend on the environment of the emitter. Overall, the research is somewhat interesting and can supply some information on the factors of the emitter orientation. However, the proposed method has been widely used for single-molecule spectroscopy, thus showing indistinctly scientific significance. Furthermore, some experimental results are inscrutable. My comments are as follows:

We thank the reviewer for their comments. However, we disagree that our work shows indistinct scientific significance over previous reports using single-molecule orientation and position imaging. In light of the comments given by reviewers #1 and #2, we were also surprised to hear that reviewer #3 found some of our experimental results "inscrutable". We address these points in more detail below where we reply to the individual comments brought up by reviewer.

1. The authors declared that the innovation of this work is the adaption and development of single-molecule defocused orientation and position imaging. However, the defocused single-molecule imaging was developed many years ago, and has been applied to investigate the orientation of organic molecules, quantum dots, and other nanomaterials, such as publications [Adv. Mater., 2009, 21, 1079; J. Phys. Chem. B, 2012, 116, 12878; ACS Nano, 2012, 6, 1268; J. Phys. Chem. Lett., 2014, 5, 2830; et al.,]. Compared with these works, what's the scientific significance of this work? What are the major problems solved by this work? Furthermore, the motivation of this work is to explore the orientation of the emitters in OLEDs. However, the emitters are pretty dense for OLEDs, far from the applied requirement of single-molecule technology. How do the authors plan to address this obstacle?

As we stated in the abstract of our manuscript and emphasised in the main text, in this work “we adapt and develop single-molecule defocused orientation and position imaging (DOPI) to measure the transition dipole orientation of individual molecules **in a state-of-the-art thermally evaporated host...**”. As we explain in our manuscript, to the best of our knowledge, there are no other previous reports of the use of DOPI (nor of any other single-molecule microscopy technique) **on thermally evaporated thin films composed of materials used in OLEDs.**

We thank the reviewer for pointing out a number of references, but we must stress that **the experiments reported in all of these were carried out on solution-processed samples involving non-semiconducting hosts.**

Nonetheless, we acknowledge that it is important to have a clear distinction between our work and the significant volume of earlier work on single-molecule studies of organic semiconductors. Indeed, it appears that in our first submission we have lost the paragraph introducing single-molecule microscopy and discussing earlier relevant work in the field. We sincerely apologize for this oversight. This paragraph has now been re-introduced as the first paragraph of page 4 of our revised manuscript and also has been expanded and thoroughly revised in response to the comments made by reviewers #1 and #3; it now reads as follows:

Single-molecule fluorescence microscopy (SMFM) methods are ideal for probing the photophysics of materials on a molecule-by-molecule basis and have contributed significantly to our understanding of biological systems, chemical reactions, and luminescent materials.²⁴ Under certain conditions and with appropriate modifications, these methods can extract information about the orientation of individual molecular TDMs.^{25–30} For example, single-molecule defocused orientation and position imaging (DOPI) has been employed to map conformational changes in biomolecules, as well as the morphology of lipid membranes and glassy polymers.^{31–35} Great strides have been made to adapt SMFM to the study of organic semiconductors, such as conjugated polymers, dendrimers, perovskite nanocrystals, and OLED emitters.^{36–45} However, those studies are mostly limited to solution-processed and solution-based systems. SMFM studies of OLEDs fabricated by thermal evaporation under high vacuum remain challenging, despite the prominence of these devices in basic research and the display industry. Furthermore, to our knowledge, SMFM has not been used to determine the orientation of the TDM of individual emitter molecules in device-relevant conditions.

Thus, as emphasised throughout our manuscript, the scientific significance of our work is that we developed a method to perform DOPI on OLED emitters in thermally evaporated, device-relevant hosts, and the major problem solved is that we have achieved such measurements in device-relevant

samples. The developments that we report—in terms of adapting DOPI measurements and in terms of the associated sample preparation—will enable other groups working on emitter orientation in OLEDs to perform these measurements.

We agree that emitter concentrations in devices are too high for single-molecule measurements. However, as we discussed in paragraph 2 of the introduction of our original submission, the influence of host-guest interactions on the orientation of emitter dopants is currently a main focus of attention in the OLED field. In this context, single-molecule measurements have unparalleled resolution for the measurement of the photophysics of emitter molecules and, due to the low concentrations of the emitter molecules, they allow discarding emitter-emitter interactions. To further clarify this point, we have added the following discussion to page 14 of our revised manuscript:

In the future, the combination of DOPI measurements of vacuum-processed samples with quantum-chemical calculations of the same systems (e.g., molecular dynamics or Monte Carlo simulations) might contribute significantly to our understanding of how these preferred orientations occur and shed light on the importance of different intermolecular interactions for the orientation of OLED emitters. The low concentration of emitters in samples used for single-molecule microscopy will allow to study the influence of host-emitter interactions on emitter orientation in the absence of artifacts originating from emitter-emitter interaction. Importantly, the comparison of these results with ensemble-averaged measurements of samples with emitter concentrations in the 1-10 wt% range can then also bring further clarification of the relevance of emitter-emitter interactions on their molecular orientation and photophysics. Moreover, the high level of control on sample architecture offered by the use of vacuum deposition and the spatial resolution provided by DOPI will help to address open questions such as the influence of interfaces—between the substrate and organic molecules, and also between different layers within an OLED—and the degree of order of the host on the orientation of emitter molecules.

2. What is the spatial resolution of the Z-axis of this system? Figure 2e presents the histogram of polar angles for emitters at different vertical positions within the film. This result is incomprehensible.

We are happy to further elaborate on the z-resolution of our measurement. Merely considering the mechanical and optical performance of our microscope, the z-stage has a minimum step size of 25 nm and the used objective (oil immersion, NA = 1.45) has an axial resolution power in terms of resolving two diffraction-limited spots on the order of hundreds of nanometres. However, the resolution of determining the z-position of molecules in our samples is not determined by these factors, but by the confidence level of our fitting routine, somewhat akin to the resolving power in super-resolution microscopy (indeed, our experiments require that no two molecules occupy the same area, i.e. that each molecule has an exclusive area of about $4 \mu\text{m}^2$). The z-position resolution comes from the fact that the defocused emission pattern of a molecule depends on the z-position of the emitter (see, e.g., *Nano Letters* 2007, 7, 7, 2043–2045 or *Optics Letters* 2003, 28, 2, 69-71) and the distance of the latter to the top surface of the film (interface with N_2).

In order to fit the z-position of each molecule, we first needed to determine the in-focus position of the sample (i.e., $dz = 0$). This was achieved by comparing the patterns from the brightest molecules to optical simulations using a modified version of the software QDControl. The selected value was then fixed for all other identified patterns in the same dataset (525 nm for the solution-processed samples, and 500 nm for the experiments for the thermally-evaporated samples).

We have estimated the error in the z-position of the molecules from the inverse-square error of each fit as:

$$\Delta z = z^{(r)} - z^{(r+l)},$$

where l indicates the l -th nearest neighbour to the best-fitted pattern $p^{(r)}$ in the library of patterns such that:

$$e_{mn}^{(r)} = \sum_{j,k} s_{jk} \left(x_{m+j,n+k} - c_{mn}^{(r)} p_{jk}^{(r)} - d_{mn}^{(r)} b_{jk} \right)^2 < c_{mn}^{(r)} \sum_{j,k} s_{jk} \left(p_{jk}^{(r)} - p_{jk}^{(r+l)} \right)^2.$$

Here, x denotes the measured pattern, c is the fitted intensity of the molecule, d is the fitted background at the position of the molecule (j,k), and e is the error in the fit. In this way, the estimated error in the z-position of the molecules satisfies the condition that the squared difference between the patterns $p^{(r)}$ and $p^{(r+l)}$ is larger than the squared difference between the best-fitted pattern and the experimental data.

To further clarify this point, we have added the following description of our error analysis to pages 2 and 3 of the revised Supplementary Information, in which we discuss our estimation of errors in general terms for the free-fitting parameters $\xi = \vartheta, \varphi, z$ as follows:

In order to fit the z-position of each molecule, we first needed to determine the in-focus position of the sample (i.e., $dz = 0$). This was achieved by comparing the patterns from the brightest molecules to optical simulations using a modified version of the software QDControl. The selected value was then fixed for all other identified patterns in the same dataset (525 nm for the solution-processed samples, and 500 nm for the experiments for the thermally-evaporated samples).

"The error in each of the free-fitting parameters ($\xi = \vartheta, \varphi, z$) was estimated from the inverse-square error of each fit as

$$\Delta \xi = \xi^{(r)} - \xi^{(r+l)} + \Delta \xi_{min},$$

where $\Delta \xi_{min}$ is the error related to having a minimum step size for each parameter in the pattern library, and l indicates the l -th-nearest neighbour to the best-fitted pattern $p^{(r)}$ in the library such that

$$e_{mn}^{(r)} = \sum_{j,k} s_{jk} \left(x_{m+j,n+k} - c_{mn}^{(r)} p_{jk}^{(r)} - d_{mn}^{(r)} b_{jk} \right)^2 < c_{mn}^{(r)} \sum_{j,k} s_{jk} \left(p_{jk}^{(r)} - p_{jk}^{(r+l)} \right)^2.$$

In this way, the estimated error in each parameter ξ satisfies the condition that the squared difference between the patterns $p^{(r)}$ and $p^{(r+l)}$ is larger than the squared difference between the best-fitted pattern and the experimental data. $\Delta \xi_{min}$ was set to the minimum step size of each parameter's grid unless $\xi^{(r+l)}$ coincided with the boundaries set for each parameter (0° and 90° for ϑ , and 0 nm and 30 nm for z in the experiments involving solution-processed samples), in which case $\Delta \xi_{min}$ was set to $\frac{1}{2}$ of the minimum grid step.

We have also added histograms of the distribution of error values and plots of such values against the weighted inverse error ($L = c/\sqrt{e}$) from all our experiments in figures S3, S4, and S5.

3. The main conclusion of this work is that "the orientation distributions strongly depend on the environment of the emitter." I think this conclusion is obvious. Can this work supply any further deep insights?

We appreciate that this conclusion appears obvious. However, it is highly relevant to the OLED community that this is pointed out, studied, and incorporated into the modelling related to emitter orientation studies.

In this work, we have introduced the framework to develop this incorporation. The translation and extension of this study to other host-emitter systems can provide long-awaited deeper insights into the molecular mechanisms controlling emitter orientation. Nonetheless, we also believe that the value of the conclusion we have been able to draw from our initial results should not be underestimated when viewed in the context of the questions currently discussed in the OLED community.

REVIEWERS' COMMENTS

Reviewer #1 (Remarks to the Author):

I thank the authors for carefully addressing all my comments. I believe the manuscript is now suitable for publication in its current form.

Reviewer #3 (Remarks to the Author):

The authors have carefully addressed all my questions. Recommend to publish!